# Two Concepts of Hepatitis B Core-Related Antigen Assay: A Highly Sensitive and Rapid Assay or an Effective Tool for Widespread Screening

**DOI:** 10.3390/v16060848

**Published:** 2024-05-26

**Authors:** Takako Inoue, Shintaro Yagi, Yasuhito Tanaka

**Affiliations:** 1Department of Clinical Laboratory Medicine, Nagoya City University Hospital, Nagoya 467-8602, Japan; clinoue@med.nagoya-cu.ac.jp; 2Research and Development Department, Advanced Life Science Institute, Inc., Hachioji 192-0031, Japan; shintaro.yagi@hugp.com; 3Department of Gastroenterology and Hepatology, Faculty of Life Sciences, Kumamoto University, Kumamoto 860-8556, Japan

**Keywords:** hepatitis B core-related antigen (HBcrAg), covalently closed circular DNA (cccDNA), HBV reactivation, point-of-care testing (POCT), mother-to-child transmission (MTCT)

## Abstract

Hepatitis B core-related antigen (HBcrAg) reflects the activity of intrahepatic covalently closed circular DNA. HBcrAg can be detected even in chronic hepatitis B patients in whom serum HBV DNA or hepatitis B surface antigen is undetectable. The HBcrAg measurement system was developed based on two concepts. One is a fully-automated and highly-sensitive HBcrAg assay (iTACT-HBcrAg) and the other is a point-of-care testing (POCT) that can be used in in resource-limited areas. iTACT-HBcrAg is an alternative to HBV DNA for monitoring HBV reactivation and predicting the development of hepatocellular carcinoma. This validated biomarker is available in routine clinical practice in Japan. Currently, international guidelines for the prevention of mother-to-child transmission recommend anti-HBV prophylaxis for pregnant women with high viral loads. However, over 95% of HBV-infected individuals live in countries where HBV DNA quantification is widely unavailable. Given this situation, a rapid and simple HBcrAg assay for POCT would be highly effective. Long-term anti-HBV therapy may have potential side effects and appropriate treatment should be provided to eligible patients. Therefore, a simple method of determining the indication for anti-HBV treatment would be ideal. This review provides up-to-date information regarding the clinical value of HBcrAg in HBV management, based on iTACT-HBcrAg or POCT.

## 1. Introduction

Because hepatitis B virus (HBV) relaxed circular DNA is continually converted to covalently closed circular DNA (cccDNA) in the nuclei of hepatocytes [1,2], lifelong follow-up to HBV infection is essential. To prevent CHB progression, patients are often treated with nucleos(t)ide analogues (NAs). The recommended endpoint of a novel therapy is a functional cure, which is defined as hepatitis B surface antigen (HBsAg) seroclearance [3,4,5].

The amount and transcriptional activity of cccDNA is also associated with the disease progression and clinical prognosis of chronic hepatitis B (CHB) patients [6]. Hepatitis B core-related antigen (HBcrAg) is a novel surrogate marker of cccDNA. It has shown a good correlation with conventional HBV biomarkers, including HBsAg and HBV DNA [6,7,8]. HBcrAg is a highly sensitive marker of the sustained transcription of cccDNA in hepatitis B e antigen (HBeAg)-negative patients, despite marked HBV DNA suppression by treatment with NAs [9]. That is, the amount of HBcrAg correlates with both the amount of HBV DNA in the blood and the amount of cccDNA in the liver [10,11]. The guideline for the treatment of CHB published by the Japan Society of Hepatology indicates that a decreased level of HBcrAg can be the goal of treatment, as described below [3]. A highly sensitive HBcrAg assay, which is about ten times more sensitive than the conventional HBcrAg assay and is fully automated, including pretreatment, was developed [12] and approved for in vitro diagnostics (IVD) by the Pharmaceuticals and Medical Devices Agency in June 2022 in Japan.

Presently, regions such as sub-Saharan Africa, Southeast Asia, and the Western Pacific are considered high-risk areas where new HBV infections occur [13]. The World Health Organization (WHO) recently issued guidance that includes options to ensure the elimination of viral hepatitis as a public health problem [14]. The WHO recommended peripartum antiviral prophylaxis for HBV-infected pregnant women with high HBV DNA levels (>200,000 IU/mL) to prevent mother-to-child transmission [15,16,17]. The guidance describes four options for country-specific goals, including the eradication of the mother-to-child transmission (MTCT) of HBV [14]. Despite the fundamental importance of measuring HBV DNA in the management of CHB, more than 95% of HBV-infected individuals live in areas where HBV DNA quantification is not readily available [18]. Given this circumstance, a rapid and simple HBcrAg assay is effective for point-of-care testing (POCT) in areas with limited resources.

Thus, current needs for HBV biomarkers include two concepts: automated, highly sensitive assay systems or rapid, simple systems that can be used as POCT in resource-limited areas. This review describes the clinical use of this new surrogate marker, HBcrAg, in the treatment of CHB or HBV reactivation, based on a highly sensitive assay (iTACT-HBcrAg), and a novel strategy of HBV prevention, based on POCT.

## 2. HBcrAg: A Surrogate Marker of Intrahepatic HBV Activity

Testing for HBcrAg was first recommended in clinical guidelines for CHB management in Japan, followed by the Asian region and then Europe [3,16,19,20]. This section introduces the unique features of HBcrAg.

### 2.1. HBV Replication Cycle

The HBV life cycle was described in detail in our previous review [13]. When HBV infects a hepatocyte, the incomplete duplex of circular genomic DNA is transferred into the nucleus and converted to cccDNA. Thus, HBV DNA persists in the nucleus as a mini-chromosome, similar to the host genome [13,21,22].

Four mRNAs are transcribed by the host system, and the HBV proteins encoded by them are translated and produced. The pre-genomic RNA (pgRNA), the template for the incomplete duplex circular DNA, and the polymerase complex, a reverse transcriptase, form a capsid covered by the HBc antigen, which is later enveloped by three types of hepatitis B surface antigens (large HBsAg, middle HBsAg, and small HBsAg) anchored in the lipid bilayer. The pgRNA is reverse transcribed by the polymerase and, in combination with the dephosphorylation of the HBc antigen, forms the infectious HBV particles (Dane particles) which are released into the blood [22,23,24].

### 2.2. HBV Biomarkers

Alternative biomarkers which reflect the cccDNA activity include the quantification of HBsAg, HBV RNA, and HBcrAg. HBsAg is also produced by the HBV genome, including from copies integrated into the host genome, and released into the blood. In contrast, because HBV RNA and HBcrAg are not produced from the integrated HBV DNA, they are more useful than HBsAg as direct biomarkers of cccDNA activity [3,19,22].

### 2.3. Components of HBcrAg

In 2002, HBcrAg was first reported as a target for the development of a sensitive enzyme immunoassay specific for circulating hepatitis B core antigen (HBcAg) and HBeAg [10], which are translated from the pgRNA, transcribed from the core promoter, and the precore mRNA, transcribed from the precore promoter (PcP) of hepatic cccDNA, respectively, [22]. HBeAg is a circulating protein, derived from the precore protein by proteolysis and secreted from the hepatocytes [25]. HBcAg is a component of the virion and forms the inner nucleocapsid that surrounds the viral DNA. Although p22cr was reported to be a component of empty particles without HBV DNA [26], it has also been reported to be found in a fraction that is not particulate, along with HBeAg [27]. The localization of p22cr is controversial, as described below [26,28]. HBeAg, HBcAg, and p22cr share a common 149 amino acid sequence (depending on the genotype) and are very similar in their monomeric structures [26,29,30]. Now, HBcAg, p22cr, and HBeAg can be measured together as HBcrAg by serological testing [31,32]. The replication cycle of HBV and components of HBcrAg are shown as Figure 1.

## 3. A High-Sensitivity HBcrAg Assay Put into Practical Use in Japan

### 3.1. Principle of the High-Sensitivity HBcrAg Assay: iTACT-HBcrAg

iTACT-HBcrAg is a highly sensitive assay with a short assay time. iTACT stands for Immunoassay for Total Antigen including Complex via pretreatment, and is an assay that analyzes the different molecular modalities present in the specimen (antigenic differences, blood molecules, and molecular complexes) [22,33].

The characteristics of the conventional HBcrAg assay (G-HBcrAg) and iTACT-HBcrAg are summarized in Table 1. In G-HBcrAg, pretreatment is performed using a conventional method, requiring 30 min of incubation. In contrast, iTACT-HBcrAg is fully automated, going from pretreatment to measurement in approximately 30 min. For iTACT-HBcrAg, the pretreatment and optimization of the reaction conditions required for the instrument resulted in a 10-fold increase in sensitivity over G-HBcrAg, and also improved specificity [12,22].

### 3.2. Basic Performance of iTACT-HBcrAg

A comparison of the values measured with iTACT-HBcrAg and G-HBcrAg for 146 HBsAg-positive sera and 105 plasma samples showed a good correlation (r = 0.99), with a regression equation slope of 1.00 in both groups [12]. Because the limit of detection (LOD) and limit of quantification (LOQ) of iTACT-HBcrAg were 1.5 Log U/mL and 1.8 Log U/mL, respectively, the measurement range was set to 2.1–7.0 Log U/mL (the instruction of use for the product [IFU]). The respective values for G-HBcrAg were 2.5 Log U/mL, 3.0 Log U/mL, and 3.0–7.0 Log U/mL (IFU), indicating that iTACT-HBcrAg increases the sensitivity and extends the measurement range [22]. The sensitivity (2.1 Log U/mL) of the iTACT-HBcrAg assay is approximately 10-fold greater than that of G-HBcrAg (2.8 Log U/mL). The dilution linearity is good through the measurement range. The dilution linearity at low values was better than that of G-HBcrAg [12].

For sera negative for HBsAg, all samples in the output range of G-HBcrAg below the LOQ (2.1–2.8 Log U/mL) were found to be below the LOQ with iTACT-HBcrAg, confirming the improved specificity. In a study using inactive HBV carriers negative for HBeAg, the positive rate was 75% for G-HBcrAg, but 97.5% for iTACT-HBcrAg, confirming the improvement in the LOQ [12].

### 3.3. Analysis of p22cr

Because HBcrAg may be detected even after HBeAg seroconversion, Kimura et al. conducted a study based on the hypothesis that HBcrAg may differ from HBcAg and HBeAg [26]. They found that p22 protein, which carries the sequence from amino acid −29 to 150 and beyond, could be isolated from the fractions containing empty particles, and named it p22cr as a major component of empty particles [22].

However, their conclusion was challenged by findings that p22cr can be detected in blood, but that p22cr (PreC) is mainly detected in the same fractions as HBeAg following density gradient ultracentrifugation and is not enriched in the empty particle fractions [28].

In addition, the phosphorylated HBcAg has been reported to be the major form of HBcAg in empty particles and RNA particles, and the phosphorylation of HBcAg is important in virion formation, such as for capsid assembly, the selective packaging of the pgRNA, and its reverse transcription [34]. The precise quantification of HBcAg [35,36], phosphorylated HBcAg ([36], our unpublished data), and HBcrAg would help further analysis of the nature of HBcrAg components.

## 4. Recent Clinical Assessments Performed Using iTACT-HBcrAg

The guideline of the Japanese Society of Hepatology recommends the use of HBcrAg as a treatment indicator, stating that “HBcrAg is an indicator of carcinogenic risk during natural history and NA treatment”, “an indicator of decreased carcinogenic risk in cases of HBeAg loss or HBsAg loss”, and “after HBV DNA becomes negative, the goal is to become HBcrAg negative” [3].

HBcrAg has been used to support the monitoring of CHB and the prediction of clinical outcomes. In this section, we briefly touch on previous reports about the role of HBcrAg and present the clinical significance of iTACT-HBcrAg (Table 2).

### 4.1. HBV Reactivation

#### 4.1.1. Relationship between HBV Reactivation and G-HBcrAg

There have been few reports on the effectiveness of G-HBcrAg for predicting HBV reactivation. In a study performed in Hong Kong on HBV-resolved patients, HBcrAg-positive patients had a significantly higher 2-year HBV reactivation rate than HBcrAg-negative patients (71.8% vs. 31%, *p* = 0.002). Serum HBcrAg positivity is a significant risk factor of HBV reactivation in HBsAg-negative, anti-HBc-positive patients undergoing high-risk immunosuppressive therapy [44].

#### 4.1.2. Usefulness of iTACT-HBcrAg in HBV Reactivation Diagnosis

Regarding the early detection of HBV reactivation, iTACT-HBcrAg is clearly superior to G-HBcrAg and fully demonstrates its true potential, and this is the most promising field. Cases with HBV reactivation have been reported in which HBcrAg is detected by iTACT-HBcrAg earlier than is HBV DNA [12,37,38]. In addition, no cases with HBV reactivation were diagnosed in patients under NA treatment whose HBcrAg was no longer detected by iTACT-HBcrAg. Based on these reports, the use of high-sensitivity HBcrAg for monitoring HBV reactivation was added as a recommendation to the guideline of the Japan Society of Hepatology [12].

The first report about iTACT-HBcrAg confirmed that the novel assay is advantageous for monitoring patients with HBV reactivation in the early phase [12]. Later, several reports have been published supporting this initial finding and demonstrating the clinical features of iTACT-HBcrAg in more detail.

Recently, the clinical usefulness of iTACT-HBcrAg was determined in patients with resolved HBV infection after NA treatment for HBV reactivation [37]. Of 25 patients with detectable iTACT-HBcrAg at the initiation of NA treatment, 17 (68%) achieved iTACT-HBcrAg loss. The recurrence of HBV reactivation after NA cessation was not observed in seven of eight patients who achieved iTACT-HBcrAg loss or became seropositive for anti-HBs during follow-up [37]. In another report, iTACT-HBcrAg was found appropriate for monitoring HBV reactivation to determine the initiation of NA treatment [38]. Of the 11 patients with undetectable HBcrAg by iTACT-HBcrAg at HBV reactivation and/or after, 10 showed unquantifiable HBV DNA.

### 4.2. Detection of Occult HBV Infection in CHB Patients with HBsAg Seroclearance

The true value of iTACT-HBcrAg can be shown in this section. In addition, the application of iTACT-HBcrAg is thought to lead to the further elucidation of CHB pathogenesis.

Wong et al. assessed the usefulness of iTACT-HBsAg and iTACT-HBcrAg assays in 96 CHB patients with HBsAg seroclearance [39]. At 10 years after HBsAg seroclearance, 20.4% and 64.5% of the patients still had detectable HBsAg and HBcrAg, respectively, and 66 (71%) of the patients had detectable HBsAg and/or HBcrAg. These results show that the iTACT assays detected a low level of HBsAg and/or HBcrAg in >70% of CHB patients, even at 10 years after seroclearance [39].

Very recently, Suzuki et al. reported that patients receiving NA treatment who achieved HBsAg seroclearance as determined by Architect HBsAg-QT (Abbott Japan LLC) rarely experienced virological relapse after NA cessation [41]. Although virological relapse occurred in 21.1% (19/90) of the patients, the HBV DNA levels of 18 patients were sustainably undetectable after temporary detection. HBsAg reversion detected by the HBsAg-QT assay (normal sensitivity) occurred in 6.7% (6/90) of patients after NA cessation. HBsAg levels according to the iTACT assay gradually decreased from the end of treatment (EOT) and in 64% (18/28) of patients reached an undetectable level at 5 years after EOT. In contrast, HBcrAg levels according to the iTACT assay slowly decreased, and in 28% (8/29) of patients reached an undetectable level at 5 years after the EOT [41].

Suzuki et al. also investigated the longitudinal profiles of iTACT-HBcrAg and iTACT-HBsAg in patients with HBsAg seroclearance (<0.05 IU/mL) [40]. The cumulative incidence rate of iTACT-HBcrAg loss after HBsAg seroclearance was higher in the inactive carrier group than in the chronic hepatitis/ cirrhosis (CH/LC) group (*p* = 0.002). Patients in the CH/LC group had higher rates of detectable iTACT-HBcrAg after HBsAg seroclearance than those in the inactive carrier group, suggesting that the presence of HBcrAg possibly contributes to CHB progression [40].

### 4.3. HBcrAg for Predicting Hepatocellular Carcinoma (HCC) Occurrence and Recurrence

The level of HBcrAg is associated with the risk of spontaneous liver carcinogenesis, especially after HBeAg seroconversion. In this subsection, we discuss the relationship between serum HBcrAg levels and HCC occurrence or recurrence in patients receiving or not receiving NA treatment.

#### 4.3.1. HCC Occurrence in Treatment-Naïve CHB Patients

It has been reported that the level of HBcrAg can be used as an indicator to stratify the risk of carcinogenesis in treatment-naïve CHB patients [45,46,47].

Using the iTACT-HBcrAg assay, a cut-off value of 2.7 Log U/mL can be set. Hosaka et al. reported that high HBcrAg levels at baseline and at one year were significantly associated with HCC development (log-rank test; *p* < 0.001) [43].

#### 4.3.2. HCC Occurrence in Treatment-Experienced CHB Patients

For treatment-experienced patients, NA reduced, but did not eliminate, the risk of HCC occurrence [48,49]. Ando et al. demonstrated that serum HBcrAg levels ≥3.4 Log U/mL are associated with progression to HCC [48]. In another report, the presence of serum HBcrAg (≥3.0 Log U/mL) for at least two years was an independent risk factor for the development of HCC [50]. In addition, regarding non-cirrhotic patients, HBcrAg > 3.90 Log U/mL predicted HCC occurrence (odds ratio [OR]: 5.95) [51].

Kaneko et al. showed that higher HBcrAg levels (cut-off value, 4.1 Log U/mL) at one year after the start of NA treatment were independent predictive factors for HCC development [52]. Hosaka et al. reported that that patients with persistently high on-treatment HBcrAg levels (cut-off values: 4.9 Log U/mL for the HBeAg-positive cohort and 4.4 Log U/mL for the HBeAg-negative cohort) were more likely to develop HCC [53]. In another report, a low but detectable level of HBcrAg (≥2.7 Log U/mL) measured by iTACT-HBcrAg potentially predicted HCC development, even if HBsAg seroclearance was achieved according to a conventional assay [42].

#### 4.3.3. HCC Recurrence in CHB Patients

HBcrAg is a predictor of the post-treatment recurrence of HCC during antiviral therapy [54]. Hosaka et al. showed that a serum HBcrAg level >4.8 Log U/mL at the time of presurgical HCC detection was related to HCC recurrence within two years after surgery [54]. Chen et al. reported that the recurrence-free survival rates were significantly lower in HCC patients with high intrahepatic cccDNA and serum HBcrAg levels [55]. Incidentally, in patients with an HCC history, low HBsAg (<3.0 log IU/mL) and high HBcrAg (>3.0 Log U/mL) values indicate a high risk of developing HBV-related HCC [56].

### 4.4. Changes of Serum HBcrAg Levels during NA Treatment

Previous studies have shown that some patients remain virologically responsive even after NA cessation. In CHB patients, HBcrAg production continues, even during highly effective NA therapy. Among patients positive or negative for HBeAg and treated with NAs, HBV DNA was undetectable in 98%, while cccDNA was still detectable in 51% [1]. Similar results have been shown in some reports [20,57,58,59].

Serum HBcrAg levels at baseline and changes during NA therapy may also serve as suitable indicators for CHB patients [11,31,60,61,62].

### 4.5. HBcrAg for Evaluating the Possibility of NA Discontinuation

Regarding the first published reports [63,64], the criteria for the discontinuation of NA was described in the Japan Society of Hepatology Guidelines for the Management of HBV Infection. The conditions required for NA discontinuation in CHB patients on NA therapy are as follows: at least 2 years after the initiation of NA therapy, serum HBV DNA at the time of discontinuation should be undetectable, and serum HBeAg should be negative at the time of discontinuation [3].

We can identify the CHB patients who do not relapse after NA discontinuation by serum HBcrAg levels. Higher levels of HBcrAg were associated with virological relapse (median, 4.9 Log U/mL [63], >3.7 Log U/mL [65]). In contrast, a lower HBcrAg level (<3.4 Log U/mL) at NAs cessation predicted non-relapse. Moreover, HBcrAg levels <3.0 Log U/mL at NAs cessation did not develop alanine transaminase (ALT) flares [64]. Hsu et al. concluded that serum HBcrAg and HBsAg levels were independent predictors of relapse after NA cessation [66].

The CREATE study group studied the predictors of HBsAg loss in a global cohort of HBeAg-negative patients with undetectable HBV DNA who had withdrawn from long-term NAs therapy [67]. Patients with low HBsAg (<100 IU/mL) and/or undetectable HBcrAg levels, particularly if non-Asian or genotype A, appear to be the best candidates for NA cessation. HBV genotype C was also independently associated with a higher probability of HBsAg loss compared to genotype B among Asian patients [67].

Very recently, Sandmann et al. evaluated the correlation of HBcrAg and HBV DNA and the predictive value for HBeAg seroconversion and HBsAg loss [68]. During NA treatment, 33% (6/18) and 9% (5/56) of patients showed HBeAg seroconversion or HBsAg loss/HBsAg <100 IU/mL, respectively. Low levels of HBcrAg before antiviral treatment were associated with these endpoints [68].

## 5. Clinical Application of the HBcrAg Assay in Resource-Limited Regions

Chemiluminescent enzyme immunoassay (CLEIA) for HBcrAg and HBV DNA quantification have limited accessibility and are expensive in resource-limited regions such as Africa. Here, we introduce recent information of a simple and unique HBcrAg assay which is useful in resource-limited regions.

### 5.1. The First-Ever Global Guidance for Country Validation of Viral Hepatitis B and C Elimination by the WHO

In 2016, the WHO called for the elimination of viral hepatitis by 2030 [69]. To eliminate HBV, it is important to promote widespread diagnosis and identify infected individuals, not only to develop breakthrough anti-HBV therapy. Currently, activities to provide HB vaccination and anti-HBV therapy are ongoing. However, following the interim follow-up results in 2019, the goal setting was reiterated.

New WHO Guidance for the country validation of hepatitis B and C elimination was released at the EASL International Liver Congress 2021 [14]. This new guidance represents the first-ever global guidance for countries seeking to validate the elimination of HBV and/or hepatitis C virus (HCV) infection as a public health problem.

Countries are encouraged to pursue the elimination of hepatitis B and C together. However, they may choose separately one of four certification options: “Option A” eliminates MTCT of HBV. “Option B” treats HCV as a public health problem. “Option C” treats HBV as a public health problem (including elimination of HBV MTCT). “Option D” eliminates HBV and HCV together as a public health problem [69].

### 5.2. The Major Focus of the New Guidance: Elimination of MTCT

In the new guidance, the control of MTCT was identified as the number one priority. In addition to postpartum vaccination, it is effective to lower the viral load in the mother’s blood to prevent MTCT. Therefore, a plan has been proposed to administer NA to HBV-infected pregnant women with high serum HBV DNA levels (≥200,000 IU/mL) [15,16,17]. Specifically, all pregnant women who come to the health center for childbirth guidance are tested for HBsAg and those who are positive are selected. Their serum HBV DNA is quantified and those who are above the standard value are selected and given medication [70].

However, in addition to being expensive, HBV DNA quantification may not be possible in the first place, or there may not be a clinical laboratory center near a health center. In both cases, it takes two weeks to one month for the measurement results of HBV DNA quantification to return. Then, an alternative indicator to HBV DNA quantification is needed. If HBV DNA quantification is not available, it has been shown to be effective to target those who are positive by HBeAg measurement (sensitivity equivalent to enzyme-linked immunosorbent assay [ELISA]) [70]. Based on these circumstances, a rapid and easy HBcrAg assay would be useful in regions with limited resources.

### 5.3. Simple Algorithm to Select CHB Patients Eligible for Anti-HBV Treatment Based on HBcrAg Assay

HBcrAg testing is inexpensive (<15 US dollars/assay, hopefully 5 US dollars/assay) and can be an alternative to diagnose clinically important HBV DNA thresholds (≥2000, ≥20,000, and ≥200,000 IU/mL). That is, HBcrAg might be an accurate alternative to HBV DNA quantification as a simple and inexpensive tool in resource-limited regions [71].

The identification of Gambian patients with CHB who had indications for anti-HBV treatment eligibility determined by the American Association for the Study of Liver Diseases (AASLD) was assessed using a new experimental algorithm that did not contain an HBV DNA assay. A simple treatment algorithm based on an HBcrAg assay alone, without an HBV DNA assay, showed a large area under the receiver operating characteristic curve (0.91 [95% confidence interval [CI]: 0.88–0.95]), with a sensitivity of 96.6% and specificity of 85.8% [71].

### 5.4. Development of HBcrAg Quantification Using Dried Blood Spots (DBSs)

To facilitate the application of HBcrAg to resource-limited regions, the use of DBSs as a tool to identify HBV-infected individuals with high viremia has been developed and assessed. With this new method, HBcrAg detection can be completed in about 5 h [72]. The LOD of HBcrAg obtained from DBSs was HBV DNA 19,115 IU/mL (4.281 Log U/mL) across the major HBV genotypes (A, B, C, D, and E). Moreover, a strong linear correlation was confirmed between DBS HBcrAg and HBV DNA levels (r = 0.94, *p* < 0.0001) in samples with high viremia (HBV DNA 3.7–7.0 Log IU/mL) [72].

### 5.5. A Rapid Detection Test of HBcrAg Using Immunochromatography

The measurement of HBcrAg requires the CLEIA system, which remains difficult to use in limited-resources regions. A HBcrAg-rapid diagnostic test (HBcrAg-RDT) based on immunochromatography has been developed and reported [73]. The reagent detects HBcrAg bound to anti-HBcrAg using an enzyme immunoassay. HBcrAg-RDT can detect high HBcrAg levels and high viremia in serum, plasma, or whole blood. Its low cost (<5 US dollars/sample), simple preparation, lack of requirement for special equipment, and short turnaround time (within one hour) favor its use in limited-resources regions [74]. A study using this reagent was conducted in The Gambia. The sensitivity and specificity of HBcrAg-RDT to diagnose HBV DNA levels were 72.7% and 91.7% for ≥2000 IU/mL, 86.7% and 88.7% for ≥20,000 IU/mL, and 91.4% and 86.3% for ≥200,000 IU/mL [74]. The procedure of HBcrAg-RDT is shown in Figure 2. If the convenient and cheaper HBcrAg-RDT (POCT) is developed, the new strategy based on POCT would be effective and provide a good cost benefit in clinical practice.

## 6. Discussion

In this review, we have discussed the future prospects for HBcrAg diagnostics. As noted above, when we think about HBV infection on a worldwide scale, there are two separate demands for HBV marker assays. One is a highly sensitive, automated assay and the other is a simple assay that can be used as POCT.

In 2022, iTACT-HBcrAg appeared in clinical practice in Japan. The advantages of iTACT-HBcrAg are that it does not need specific skills, and it is less expensive than serum HBV DNA assays. Moreover, iTACT-HBcrAg has a more rapid turnaround time than the conventional HBcrAg assay, with results obtainable within 30 min, rather than the approximately 7 h required for serum HBV DNA assays [12]. In addition, because the entire process is fully automated, the results can be obtained in a shorter time. Not only does this reduce the burden on laboratories, it is also expected to reduce the burden on patients by enabling in-hospital testing prior to routine medical examinations.

HBcrAg has shown the potential of expanding its application to clinical cases that have been difficult to discriminate and stratify with conventional reagents. In addition to Japan, Taiwan, Hong Kong, and Europe have reported the clinical usefulness of HBcrAg levels, determined using G-HBcrAg [6]. Some have used reference values outside the measurement range of previous reagents (G-HBcrAg). As discussed above, iTACT-HBcrAg shows a high correlation with G-HBcrAg and has been confirmed to have improved dilution linearity and specificity at lower values compared to G-HBcrAg [12]. Therefore, with regard to the utility demonstrated in studies that set the cut-off lower than the LOQ due to the lack of sensitivity of G-HBcrAg, it is necessary to reexamine this issue using iTACT-HBcrAg.

In contrast, the simplified test is being applied to approaches to achieving the 2030 elimination goals. The development of POCT for more convenient and accurate HBV markers is aimed at the efficient identification of individuals infected with HBV. The system uses DBSs and immunochromatography for HBcrAg and has shown promise for identifying HBV-infected patients with high viremia who need anti-HBV therapy. In locations like Africa, where the medical institution and home often are far apart, we must consider the possibility that once a person leaves the medical institution, he/she may never return. In this respect, the HBcrAg-RDT reported recently makes sense and is an excellent technique consistent with local requirements [75].

The WHO recommends that all of the pregnant women undergo screening for HBsAg, along with human immunodeficiency virus (HIV) testing [17]. If HBsAg is positive, the serum HBV DNA should be quantified. If the level is ≥5.3 Log U/mL, peripartum antiviral prophylaxis with tenofovir disoproxil fumarate (TDF) is recommended from the 28th week of pregnancy to delivery [15,76]. Peripartum antiviral prophylaxis can be discontinued either immediately after delivery or from 12 weeks postpartum, depending on the specific guidelines [15,75,76]. In all international guidelines, long-term anti-HBV treatment is recommended in cases of ongoing viral replication associated with moderate or severe liver inflammation and/or fibrosis [15,16,75,77]. The assessment of liver diseases in HBsAg-positive women during pregnancy may present a great opportunity to recognize those who may benefit from long-term anti-HBV treatment [15,77].

However, using these simplified criteria may result in the overtreatment of a large proportion of pregnant women infected with HBV. Currently, HBV treatment is long-term and associated with potential side effects, including renal damage. Therefore, it would be ideal if a simple method could be used to determine the indication for anti-HBV treatment. Overall, international guidelines are inconsistent and ambiguous due to their complexity and different set points [78]. Future WHO guidelines should be revised to simplify and standardize the indications for anti-HBV therapy and to ensure appropriate approaches to testing and treatment.

## Figures and Tables

**Figure 1 viruses-16-00848-f001:**
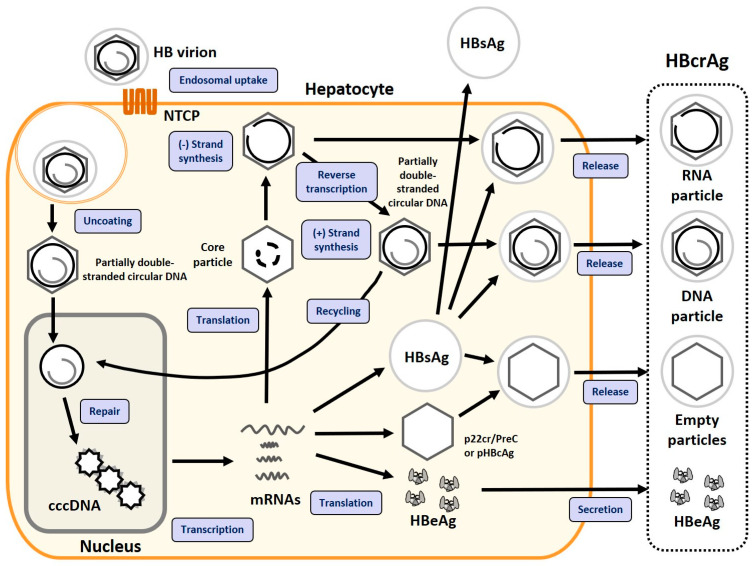
The replication cycle of HBV and HBV biomarkers, including HBcrAg. This diagram schematically shows the HBV replication cycle, biomarkers that serve as indicators of HBV activity, and the production mechanism of each antigen. This figure was created by modifying a figure previously published in our report [6]. Abbreviations: HBV, hepatitis B virus; cccDNA, covalently closed circular DNA; HBcrAg, hepatitis B core-related antigen; HBeAg, hepatitis B envelope antigen; HBcAg, hepatitis B core antigen; pgRNA, pre-genomic RNA; NTCP, sodium taurocholate cotransporting polypeptide.

**Figure 2 viruses-16-00848-f002:**
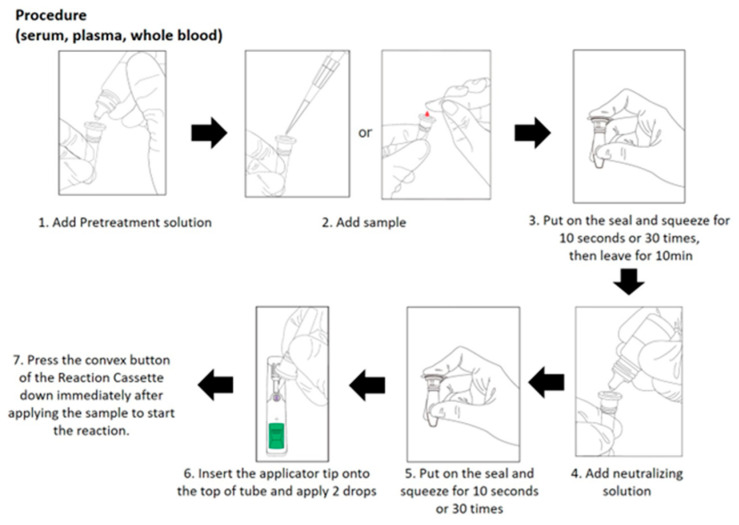
HBcrAg detection procedure by ESPLINE HBcrAg (RUO) [22]. This figure is based on Figure 4 in “Fundamental performance and clinical utilities of LumipulsePresto^®^ iTACT^®^ HBcrAg, a novel highly sensitive immunoassay for hepatitis B core-related antigen” by Yagi [22]. RUO is for the detection of HBcrAg in serum, plasma, whole blood, or dried blood spots. Abbreviations: HBcrAg, hepatitis B core-related antigen.

**Table 1 viruses-16-00848-t001:** Comparison of the conventional and highly sensitive HBcrAg assay [22].

	Conventional HBcrAg Assay	Highly Sensitive HBcrAg Assay
Reagent name	G-HBcrAg (LUMIPULSE^®^ HBcrAg)	iTACT-HBcrAg (LUMIPULSE Presto^®^ iTACT^®^ HBcrAg)
Available specimens	Serum or plasma	Serum or plasma
Pre-treatment	Utilization of manual procedure	Fully automatic
Antibodies to detect	Solid phase: 3 antibodies Labeled: 2 antibodies	Solid phase: 4 antibodies Labeled: 2 antibodies
Sample volume	During off-board pretreatment: 150 μL (at the time of measurement: 60 μL)	50 μL
Measurement Time	60 min. (30 min. for pretreatment +30 min. for measurement)	33 min. (6 min. for pretreatment +27 min. for measurement)
Measurement range	3.0–7.0 Log U/mL	2.1–7.0 Log U/mL
Valid period of on-board calibration curve	30 days	90 days

Abbreviations: HBcrAg, hepatitis B core-related antigen; iTACT-HBcrAg, high-sensitivity HBcrAg assay.

**Table 2 viruses-16-00848-t002:** Previous reports regarding the efficacy of iTACT-HBcrAg.

Target	Patients	Findings	HBcrAg	References
HBV reactivation	n = 13	Nine and two of thirteen HBV-reactivated patients were HBcrAg-positive by iTACT-HBcrAg before and at HBV DNA positivity, respectively.	2.1 Log U/mL	[12]
	n = 27	iTACT-HBcrAg became negative in 68% (17/25) after NA treatment. Eight patients who achieved iTACT-HBcrAg loss or anti-HBs seropositivity had no recurrence of HBV reactivation after NA discontinuation, except for one patient who did not have anti-HBs after allogeneic transplantation.	2.0 Log U/mL	[37]
	n = 44	HBcrAg was detectable by iTACT-HBcrAg before HBV DNA was quantifiable in 15 of the 27 patients. Of the 11 patients with HBV reactivation and undetectable HBcrAg by iTACT-HBcrAg at HBV reactivation and/or thereafter, 10 had unquantifiable HBV DNA and none developed hepatitis.	2.0 Log U/mL	[38]
CHB patients with HBsAg seroclearance	n = 96	Using iTACT-HBcrAg, HBcrAg was detectable in 78.3% and 65.9% of samples collected before and after SC, respectively. At 10 years after SC, 64.5% of the patients still had detectable HBcrAg.	2.1 Log U/mL	[39]
	n = 120	The proportion of patients who lost HBcrAg first was significantly higher in the IC group (*p* = 0.008). The cumulative incidence of iTACT-HBcrAg loss after HBsAg SC was higher in the IC group than in the CH/LC group (*p* = 0.002).	2.0 Log U/mL	[40]
CHB patients with HBsAg seroclearance by NA treatment	n = 90	HBcrAg levels by iTACT assay slowly decreased, and in eight of twenty-nine patients (28%) reached an undetectable level at 5 years after EOT.	2.1 Log U/mL	[41]
CHB patients negative for HBeAg	n = 161	HBcrAg was detectable in the sera of 97.5% (157/161) of patients with CHB by iTACT-HBcrAg, of whom 75.2% (121/161) had ≥2.8 Log U/mL HBcrAg and 22.4% (36/161) had 2.1–2.8 Log U/mL HBcrAg, which was undetectable by G-HBcrAg.	2.1 Log U/mL	[12]
HCC development	n = 17	HBcrAg ≥2.7 Log U/mL in five patients with HCC after HBsAg seroclearance was significantly higher than the level of HBcrAg in 12 patients who did not develop HCC (100% [5/5] versus 33% [4/12], *p* = 0.029).	≥2.7 Log U/mL	[42]
	n = 180	In 110 patients (61.1%) with ≥4.0 Log U/mL at baseline (high HBcrAg cohort), HBcrAg declined to ≤2.9 Log U/mL at year 1 in 25 patients (22.7%). The adjusted hazard ratio for HCC incidence was significantly lower in patients with HBcrAg ≤ 2.9 Log U/mL at year 1 than in those in the high HBcrAg cohort.	>2.9 Log U/mL at 1 year	[43]

Abbreviations: HBcrAg, hepatitis B core-related antigen; iTACT-HBcrAg, high-sensitivity HBcrAg assay; HBV, hepatitis B virus; NA, nucleos(t)ide analogue; CHB, chronic hepatitis B; HCC, hepatocellular carcinoma; HBsAg, hepatitis B surface antigen; IC, inactive carrier; SC, seroclearance; CH, chronic hepatitis; LC, cirrhosis; EOT, end of treatment; HBeAg, hepatitis B e antigen.

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
