# Peer review of "Two Concepts of Hepatitis B Core-Related Antigen Assay: A Highly Sensitive and Rapid Assay or an Effective Tool for Widespread Screening"

_viruses, 2024, doi:10.3390/v16060848_

Round 1
Reviewer 1 Report
Comments and Suggestions for Authors
HBcrAg is a marker recently introduced in the monitoring of HBV chronic infection. The main characteristic of this marker is its correlation with cccDNA level.
The review is interesting but there are some point that need to be revised to improve the paper.
Paragraph 2.1 “HBV replication cycle”: HBV replication cycle should be better describe. In the present form, only transcription phase has been reported.
Paragraph 2.2 “HBV biomarkers”: This paragraph should include only biomarkers, the first phrase about NAs and their action, shouldd be eliminated. All markers and their significance should be described.
Paraghaph 3.1: Can the authors better describe the principle of iTACT-HBcrAg? Why does this assay show a high sensitivity compared to conventional HBcr Ag? A description of this assay is given in Section 5.5. In my opinion, better organization of these paragraphs is needed.
I believe that there are too many sections and subsections and some sections are very short and with no discussion of the data reported.
A reorganizing of the structure could improve the quality of the review
Author Response
Reviewer 1
Comments to the Authors:
HBcrAg is a marker recently introduced in the monitoring of HBV chronic infection. The main characteristic of this marker is its correlation with cccDNA level.
The review is interesting but there are some point [sic] that need to be revised to improve the paper.
Response: We appreciate your important advice to improve our review.
Paragraph 2.1 “HBV replication cycle”: HBV replication cycle should be better describe [sic]. In the present form, only transcription phase has been reported.
Response: We thank you for your valuable comment. Our previous review (Inoue T and Tanaka Y, Hepatitis B Virus and Its Sexually Transmitted Infection – an Update. Microbial Cell 2016, 3 (9): 420-437) has described the HBV life cycle in detail. We introduced the review as a reference (line 74).
Paragraph 2.2 “HBV biomarkers”: This paragraph should include only biomarkers, the first phrase about NAs and their action, shouldd [sic] be eliminated. All markers and their significance should be described.
Response: We thank you for the comment which improves our review. In subsection “2. 2. HBV biomarkers”, the first paragraph about nucleos(t)ide analogues and their action has been deleted.
Paraghaph [sic] 3.1: Can the authors better describe the principle of iTACT-HBcrAg? Why does this assay show a high sensitivity compared to conventional HBcr Ag [sic]? A description of this assay is given in Section 5.5. In my opinion, better organization of these paragraphs is needed.
I believe that there are too many sections and subsections and some sections are very short and with no discussion of the data reported.
A reorganizing of the structure could improve the quality of the review.
Response: We thank you for your insightful comment. In this review, we describe about current needs for HBV biomarkers by two concepts. Section 3 is for the clinical use of HBcrAg based on a highly sensitive assay (iTACT-HBcrAg). Section 5 is for a novel strategy of HBV prevention, based on POCT. We prefer that these two concepts should be mentioned in separate paragraphs.

Reviewer 2 Report
Comments and Suggestions for Authors
The authors reviewed the clinical evidence of HBcrAg, including the iTACT method, and its usefulness in countries where serum HBV-DNA level cannot be easily measured.
This paper covers a wide range of topics related to HBcrAg and will be instructive to readers and provide new insights.
However, some minor points need to be corrected.
1) The way in which "cut-off value" are written should be standardized. For example, you described "cutoff value" in line 258, but "cut-off values" line 261.
2) An inequality sign is required before ”3.0 Log U/mL" on line 273. {ex. HBcrAg (>3.0 Log U/mL)}
Comments on the Quality of English LanguageThis manuscript needs minor revisions, but English is almost fine.
Author Response
Reviewer 2
Comments to the Authors:
The authors reviewed the clinical evidence of HBcrAg, including the iTACT method, and its usefulness in countries where serum HBV-DNA level cannot be easily measured.
This paper covers a wide range of topics related to HBcrAg and will be instructive to readers and provide new insights.
However, some minor points need to be corrected.
Response: We thank you for your insightful comment, which improve our manuscript.
The way in which "cut-off value" are written should be standardized. For example, you described "cutoff value" in line 258, but "cut-off values" line 261.
Response: We thank you for pointing out. Based on your advice, we have unified the description "cut-off value"(line 257).
An inequality sign is required before ”3.0 Log U/mL" on line 273. {ex. HBcrAg (>3.0 Log U/mL)}
Response: We thank you for pointing out our mistake. An inequality sign is required before ”3.0 Log U/mL" on line 272.

Round 2
Reviewer 1 Report
Comments and Suggestions for Authors
The authors partially satisfied my requests, but explained to me why their review was structured as reported.